# Assessing the Needs of Those Who Serve the Underserved: A Qualitative Study among US Oncology Clinicians

**DOI:** 10.3390/cancers15133311

**Published:** 2023-06-23

**Authors:** Manali I. Patel, Leslie Hinyard, Fay J. Hlubocky, Janette K. Merrill, Kimberly T. Smith, Sailaja Kamaraju, Daniel Carrizosa, Tricia Kalwar, Lola Fashoyin-Aje, Scarlett L. Gomez, Sanford Jeames, Narjust Florez, Sheetal M. Kircher, William D. Tap

**Affiliations:** 1Department of Medicine, Stanford University School of Medicine, Stanford, CA 94305, USA; 2Medical Services, VA Palo Alto Health Care System, Palo Alto, CA 94304, USA; 3Department of Health and Clinical Outcomes Research, Saint Louis University School of Medicine, St. Louis, MO 63104, USA; leslie.hinyard@health.slu.edu; 4Department of Medicine, University of Chicago School of Medicine, Chicago, IL 60637, USA; fhlubock@medicine.bsd.uchicago.edu; 5American Society of Clinical Oncology, Alexandria, VA 22314, USA; janette.merrill@asco.org (J.K.M.); kim.smith@asco.org (K.T.S.); 6Department of Medicine, Medical College of Wisconsin, Milwaukee, WI 53226, USA; skamaraju@mcw.edu; 7Levine Cancer Institute, Charlotte, NC 28204, USA; 8Medical Services, Veterans Administration, Miami Healthcare System, Miami, FL 33125, USA; tricia.kalwar@va.gov; 9US Food and Drug Administration, Silver Spring, MD 20993, USA; 10Department of Epidemiology, University of California—San Francisco School of Medicine, San Francisco, CA 93701, USA; 11Department of Social and Behavioral Sciences, Huston Tillotson University College of Arts and Sciences, Austin, TX 78702, USA; sejeames@htu.edu; 12Department of Medicine, Dana-Farber Cancer Institute, Harvard Medical School, Boston, MA 02115, USA; narjust_duma@dfci.harvard.edu; 13Department of Medicine, Robert H. Lurie Comprehensive Cancer Center, Northwestern University, Chicago, IL 60611, USA; sheetal.kircher@nm.org; 14Department of Medicine, Memorial Sloan Kettering Cancer Center, New York, NY 10065, USA; tapw@mskcc.org

**Keywords:** cancer care disparities, health equity, evidence-based cancer care delivery

## Abstract

**Simple Summary:**

Overcoming cancer health disparities requires an understanding of the etiologies that drive these persistent disparities. To date, little knowledge exists regarding barriers and facilitators experienced by clinicians in their delivery of cancer care for populations most at risk for cancer disparities, such as low-income and racially and ethnically minoritized populations. The aim of this study was to assess the perspectives of clinicians across the United States, with a particular emphasis on understanding modifiable barriers to ensuring evidence-based cancer care delivery for populations most at risk for cancer disparities. Findings revealed the impact of clinical infrastructure, resources, and support to deliver cancer care equitably, social and economic challenges that often inhibit evidence-based care delivery, and the crucial importance of relationships with the community and other clinicians in the community, as well as attention to clinician wellness. These findings reveal areas of unmet need ripe for solutions to achieve cancer health equity.

**Abstract:**

Background: The American Society of Clinical Oncology established the ‘Supporting Providers Serving the Underserved’ (SUS) Task Force with a goal to develop recommendations to support cancer clinicians who deliver care for populations at risk for cancer disparities. As a first step, the Task Force explored barriers and facilitators to equitable cancer care delivery. Methods: Clinicians across the United States who deliver care predominantly for low-income and racially and ethnically minoritized populations were identified based on lists generated by the Task Force and the Health Equity Committee. Through purposive sampling based on geographical location, clinicians were invited to participate in 30-60 min semi-structured interviews to explore experiences, barriers, and facilitators in their delivery of cancer care. Interviews were recorded, transcribed, imported into qualitative data management software, and analyzed using thematic analysis. Results: Thematic analysis revealed three major themes regarding barriers (lack of executive leadership recognition of resources; patient-related socio-economic needs; clinician burnout) and two major themes regarding facilitators (provider commitment, experiential training). Conclusions: Findings reveal modifiable barriers and potential solutions to facilitate equitable cancer care delivery for populations at risk for cancer disparities.

## 1. Introduction

Over the past decade, clinical advances have improved cancer survival, yet morbidity and mortality disparities persist [1,2,3]. These disparities are most pronounced among populations that have historically received inadequate health care and health care services, such as low-income and racially and ethnically minoritized populations. Across the United States, oncology clinicians deliver care for these populations who experience cancer disparities due to structural racism, suboptimal access to healthcare (e.g., due to lack of health insurance, underinsured, geographical isolation), and other causes [3,4]. While studies have addressed the unmet needs among clinicians in the provision of cancer screening [5,6,7] and studies examining the multidisciplinary care for patients outside the United States [8], limited data exists regarding the barriers and facilitators that oncology clinicians within the United States experience in delivering cancer care among populations most at risk for cancer disparities.

The American Society of Clinical Oncology (ASCO) strives to support oncology clinicians and other professionals caring for people with cancer. ASCO is committed to addressing cancer disparities and has launched a multipronged approach to achieve cancer health equity [3,9]. Through ASCO’s Health Equity Committee (HEC), the Society has advanced several initiatives to eliminate cancer health disparities, including the establishment in 2019 of the “Supporting Providers Serving the Underserved (SUS)” Task Force. The SUS Task Force, composed of oncologists and other cancer professionals with health services and cancer health equity expertise, was created with the goal of developing recommendations to engage, support, and learn from oncology clinicians who deliver care predominantly for populations most at risk for cancer disparities. The SUS Task Force chose to focus on low-income and racially and ethnically minoritized populations given the prevalence of cancer disparities among these groups. As a first step, the SUS Task Force conducted this qualitative study given the limited data available to explore barriers and facilitators to equitable cancer care delivery from the perspective of clinicians. The qualitative study would then inform a quantitative survey that would be distributed as a second step among oncology clinicians nationally to generate data on a national level that could inform recommendations and solutions.

## 2. Materials and Methods

The SPUS Task Force utilized the definition of “underserved population’’ as described by the Health Resource and Services Administration (HRSA), which defines this group as a population of individuals, specifically low-income and racially and ethnically minoritized populations, who have historically received inadequate health care and health care services [10]. The SUS Task Force, in collaboration with prior and current Health Equity Committee (HEC) members, generated a list of 55 medical oncologists across the United States who, based on their knowledge and expertise, spent greater than 25% of their time treating or having had experience treating predominantly low-income and racial and ethnic minority populations with cancer in their daily clinical practice. The list included clinicians who applied to ASCO’s grant programs that focused on serving underserved patients.

A total of 12–15 participants were anticipated to reach thematic saturation. However, as this study was conducted during the height of the COVID-19 pandemic and given the time limitations of clinicians targeted for study participation, the Task Force sent email invitations inviting 24 clinicians to participate in 30–60 min virtual, semi-structured interviews. These 24 clinicians were purposefully selected from the generated lists of 50 clinicians to participate in interviews based on geographical location. Participants provided verbal informed consent upon initiation of the interview and completed a survey prior to the interview to define their demographic characteristics and to describe the demographic and clinical characteristics of the patient population for whom they deliver care.

The interviews were conducted by video conferencing via a web-based platform between 1 March 2021 and 30 April 2021 by investigators (JM and KS) using a semi-structured interview guide developed by members of the Task Force and created in discussion with members of the HEC (Appendix A). The interview questions were developed using the Socio-Ecological Framework [11]. Interview questions explored clinician experiences delivering cancer care for low-income and racial and ethnic minority populations and approaches that could support their care delivery. Two non-clinical members of the team experienced in qualitative methodology, interviewing techniques, and public health (JK and KS) conducted all interviews to limit the impact of clinician-researcher assumptions and experiences on participants’ responses. There were no prior relationships or interactions between any of the research team members and the participants. Interviews were recorded, transcribed, stripped of potential identifiers, and imported into qualitative data management software (NVIvo version 12).

Descriptive statistics were calculated for demographic and clinical practice characteristics, which included frequencies for categorical variables. The analysis was performed using thematic analysis as described by Braun and Clark [12], given the limited research on the topic. The flexibility of this approach permitted an understanding of the experiences of participants in the broader context of their roles in delivering cancer care for underserved populations while remaining focused on the data and recognizing practical and realistic limitations [12]. The perspectives of the authors were ontological (themes shaped by participants’ experiences and reflecting the realities of both participants and researchers) and methodology (shaped by experiences in data collection and analysis).

Three investigators with experience in qualitative methodology and backgrounds in public health, anthropology, oncology, psychology, ethics, social work, and health services research (MP, LH, and FH) read samples of the transcribed text to get a broad understanding of the data content. The investigators used a mixed inductive-deductive iterative approach in which the investigators identified constructs driven directly by the data and applied a socioecological framework to the data to generate codes for key points and a codebook. Full transcripts were coded independently by three experienced qualitative coders (as part of Principal Investigator MP’s health services research laboratory) who consecutively coded full transcripts, discussed discrepancies, and modified the codebook with the lead investigator (MP). A Cohen’s kappa was calculated to measure coder consistency using all quotes from major code categories, with scores that ranged from 93 to 97% suggesting excellent consistency [12,13,14]. Thematic analysis of 1582 unique quotations was conducted, and codes were sorted into broader themes to ensure that data with themes were consistent and that themes were distinct and fine-tuned until thematic saturation was reached, defined as a point when no new additional information was found in the data [12,13,14,15]. Discussions were held between coders and the research team throughout the analysis phase to reflect on how personal experiences were used to attach meaning to themes and if assigned meanings resonated across team members from different clinical and research backgrounds. The research team held reflexive team discussions to ensure meanings assigned to themes were consistently agreed upon by all team members, as described by Barry et al. [16]. This process allowed the research team to identify personal paradigms regarding research involving underserved populations, express an orientation to qualitative research, negotiate a research methodology aligned with the aims of the study, and work to reduce bias in interpretation. Triangulation was applied to compare data gathered from discussions with experts held with other HEC members prior to the study’s start with those gathered from the semi-structured interview participants [17]. Analytical memos, field notes, a codebook, coding rules, and meeting notes were kept as an audit trail for dependability and confirmability [18]. The study was reviewed and approved by the Stanford University Institutional Review Board.

## 3. Results

Of the 24 eligible participants invited to participate, 12 participated; 10 (2.4%) did not respond to the initial invitation, and 2 (8.3%) agreed to participate but did not have available time to participate in the interviews. Table 1 shows the demographic characteristics of the participants. There were an equal number of male (*n* = 6; 50%) and female (*n* = 6; 50%) respondents. Participants self-identified as African American or Black (*n* = 4, 33%); Asian American Native Hawaiian and other Pacific Islanders (*n* = 2, 17%); or White (*n* = 5; 42%); 1 (8%) participant preferred to not answer. Most participants had completed their terminal degree more than 20 years ago (*n* = 5; 42%) and spent >25% of their clinical effort delivering care for low-income and racial and ethnic minority populations (*n* = 9; 74%). Most (*n* = 8; 67%) lived in the community where they delivered care and were employed by a hospital or health-system-owned practice, group, or department (*n* = 5; 42%) with 10 or less full-time-equivalent oncology clinicians and subspecialists (*n* = 4; 33%). Most (*n* = 10; 83%) reported that their patient population was reflective of the broader community in which they practiced and reported that most of their patients were insured by public insurance, either Medicare or Medicare Advantage (35%) or Medicaid (30%), with no insurance or other forms of payment (such as charity care) representing 18% of their patient population. Most participants reported that Non-Hispanic White patients represented 32% of their population, followed by Black or African American (23%), Hispanic or Latino (19%), and Asian American Native Hawaiian and other Pacific Islanders (15%).

Thematic analysis revealed three major themes regarding barriers (Table 2) and two major themes (Table 3) regarding facilitators as it pertains to cancer care delivery for low-income and racial and ethnic minority populations with cancer.

## 4. Barriers

### 4.1. Theme 1: Lack of Executive Leadership Recognition of Resources Required

Participants noted the significant amount of time and resources needed to deliver evidence-based cancer care and the continual need to justify necessary resources to executive leaders. For example, one explained, “..the administration [needs to] understand that we need a staff person just to coordinate records. [It] doesn’t make sense to them because they’re used to a surgery clinic or a primary care clinic where a patient comes in, sees the doctor, goes on about their day”. Another noted, “I don’t think we are getting enough support from the outside, so we have to survive with whatever we have”. One stated, “it all boils down to having enough resources to make these things to make these changes that are unique to this population”.

Within the primary theme of lack of administrative support and resources, two additional subthemes emerged.

### 4.2. Subtheme 1: Reliance on Short-Staffed Team Members

All participants described dependence on under-resourced and understaffed teams to address health-related social needs and noted the significant need for dedicated staff to ensure evidence-based care delivery. One stated, “We have quite a busy practice. We are only two oncologists here. We don’t have enough navigators to help [patients]. We don’t have enough social workers to be assigned to patients”. Another noted, “We don’t even have a dedicated social worker here. The patient navigator, she does the social work. Whenever we are faced with transportation challenges or meals on wheels and all that, we ask her if the patient is eligible”. Another stated, “From the moment a patient is referred to the time they start treatment…to coordinate [care] is really hard. I want to hire a navigator, but we just haven’t been able to. We’ve had an opening for over a year and haven’t been able to fill it”. One stated, “Interestingly, less-resourced clinics, like ours, do a phenomenal job, considering how difficult it is to generate resources for people who may not be able to pay. In other words, anybody can come here and if they don’t have resources, they’re not turned away. [They can be turned away] at a private, more-resourced hospital, but it doesn’t happen here. That is the challenge. Then, all the frustrations come out of that fact, that we need more resources and [the leaders] are not investing”.

### 4.3. Subtheme 2: Constant Need to Prove Value

All participants noted unmet needs in advocating for additional resources and support staff to ensure evidence-based cancer care delivery. One stated, “So me talking to the administration, they don’t get it. They don’t understand what it takes. What I constantly get told is we don’t need the staff to do that, which I hate to think is the barrier”. Another stated, “So people want to restrict the lens of what they look at to a convenient sample when, in reality, every life matters. So, the fact that somebody didn’t come into your center of excellence doesn’t mean that their life is not valuable to someone, doesn’t mean that you shouldn’t be counting them when you’re trying to figure out how well are you doing and where the opportunity is to do better. In fact, it is looking at those places that gives you the greatest opportunity to do better and have greater impact than if we just wanted to keep buffing and shining the glittering part of our healthcare system that we’re most proud of”. Another stated, “We need [additional navigators and social workers], but we don’t have them, and will the hospital invest in it, why? You have to keep asking for certain things.. And you get X amount. So, I feel like it’s a big issue”. One stated, “The urban poor have not traditionally been the focus on my healthcare system, and so we have to explain and show time and time again why and what is needed. It’s exhausting”.

### 4.4. Theme 2: Care Delivery Inhibited by Unmet Complicated Social and Economic Needs

Participants noted that the delivery of evidence-based care was adversely impacted by challenging patient social and economic needs. Many discussed how the healthcare system currently does not address these needs or only partially addresses them. One stated, “The [health] system is not set up to do this… I can’t get care to my patients if they do not have a way to get to my clinic or they do not have a way to eat or they do not have a place to live. This is not something that our clinic is set up to do”.

All participants noted that health-related social needs impacted patient wellbeing and frequently inhibited evidence-based care delivery. Many noted that patients lacked transportation to the clinic or had extensive commutes. Almost all participants, however, had identified solutions to overcome transportation-related issues. One stated, “Transportation is a concern, but we have ways to get people the rides they need”. All participants noted the prevalence of other health-related social needs such as food insecurity, housing instability, and limited health literacy. One participant stated, “There were patients that would use their rideshare to get to the clinic because that was the one thing that was covered, but they were hungry”. Another stated that “patients would forego treatment with a response of ‘I don’t have money for groceries.’” Housing instability was also noted as particularly concerning and impacting care delivery. One participant stated, “The social circumstances are something that we cannot help. If someone has no place to stay and they stay in a shelter, their capacity to come back and forth is not possible. Somebody who is living in a shelter, then there’s nobody at the other end to take care of the poor in a safe manner. These are some of the things that are beyond individual’s control and definitely beyond the control of the hospital”.

Issues involving health literacy were frequently mentioned as barriers to care. “I’m using words like ‘doo-doo hole’ to explain anal canal to a patient because that’s the only word that they would really understand. The challenge is identifying the patients who require that terminology and require that discussion”. Another stated, “So my thoughts are that there may be challenges with limited health literacy not just in our non-English-speaking population but throughout our population”.

In response to challenges regarding health-related social needs, participants indicated searching for solutions to ensure continuity of care for their patients. Participants described how they created transportation solutions, such as ride shares, clinic vans, and gift cards for gas money, given the prevalence of this unmet need among their patient population. One stated, “I have patients who live in a place where they don’t have personal transportation means. They have to depend on a Medicare/Medicaid van to bring them to get treatment or to be seen, and what have you”. Another participant addressed how the clinic provides monetary support for both transportation and food. “We give a lot of gas vouchers to our patients to give credits for that. We do a lot of gas vouchers, meals, meal vouchers as well. This is not something that happens rarely. I mean, every day we have patients who are going to get one or the other or both”. Some described how they responded to these unmet social needs by changing their clinical workflows. One stated, “What we’ve done, at least in my practice, is that we don’t let them walk out until they have their appointment and until they have their imaging scan scheduled. We wrap up all of it in the office. It’s resource intensive for my medical assistants and the navigators, but that’s the only way we can ensure that continuity of care”.

### 4.5. Theme 3: Burnout Prevalent Due to Lack of Resources and Time Spent Advocating and Proving Value

All participants noted that a lack of resources to deliver equitable, evidence-based cancer care coupled with increasingly burdensome administrative processes negatively impacted their wellbeing. Many noted that these issues contributed greatly to “burnout”. One stated, “So, I would say, my sources of frustration are more systemic. I mean, it affects all of our patients and all of our doctors. Preauthorization, documentation. I’m sure a lot of this was meant [to help]. I don’t know about preauthorization, but some of the documentation was meant to make things more efficient but if I made a little graph of my practice of how much time I spend with patients and how much time I spend doing paperwork, I’m sure those lines are becoming inverse”. One stated, “Now I understand the price that we often pay as professionals. We feel rewarded by the opportunity to serve others. But oftentimes, we steal time from families and our loved ones. There are only 24 h through any day, and I was alone in my work of filling out paperwork and more paperwork just to get the care to the people who need it”.

Within the primary theme of burnout, two additional subthemes emerged.

### 4.6. Subtheme 1: Wellbeing of Clinicians Associated with Delivery of Evidence-Based Care 

Many participants noted that burnout was associated with their inability as professionals to overcome their patients’ social and economic needs—needs that directly inhibited the delivery of evidence-based care. One stated, “So the disparity frustration I’ve really struggled with like, how am I going to get the patient to see a gastroenterologist, a pulmonologist, who will take them happily without supplementing and pleading and requesting them get squeezed in to do it because they have no insurance”. Another stated, “[Our patients] have a significant comorbid health load, and they have significant challenges about what they can’t meet with transportation or not having a working phone, being homeless, things that we see all the time. I think that really impacts the way providers try to then deliver the care and then manage their own time and affects us a lot”. Another stated, “If you don’t have social workers, navigators, good systems in place, then it’s very easy to work your clinic load then round at a hospital and not finish your day until 10:00 or 11:00 at night”. Another stated, “There are only a few of us. So, we would work through lunch, and it’s just not sustainable; you have to use time and manage your time in ways that is still going to protect your ability to not faint, your ability to be appropriate with patients; if you don’t have anything left, you don’t have anything to give to your patients”.

### 4.7. Subtheme 2: Burnout from Worry about Other Team Members

Participants noted their concern about the wellbeing of their team members, including nurses, social workers, and ancillary staff members. One stated, “Yeah, burnout is real. I’ve remained cognizant of that, but it affects everyone in the care team. So it’s not just the oncologist, it’s everyone in the care team. So I think I recognize it in nurses. I think I recognize it in clinical staff, support staff. When you’re dealing with very sick, high-acuity patients with a lot of needs, burnout problem is real”. One stated, “We don’t go into this thinking that we go on to fix problems right away. So, we get used to that; psychologically, we have to have some amount of resilience in order to survive. Even as there is progress, it is still a challenging discipline that requires resilience on the part of the physician or the patient care providers. It can be a lot to think about how nurses and others are experiencing the brunt”.

### 4.8. Facilitators

While barriers to care delivery were stressed throughout the interviews, participants also noted facilitators.

### 4.9. Theme 1: Local Connections with Community Partners and Foundations

Participants emphasized the importance of forming collaborative relationships with community members and community organizations to facilitate cancer care delivery. One stated, “Without the [community organizations], we would just be in dire, dire straits because so many patients wouldn’t be able to get the care they need”. Another described how collaboration with community organizations helped them to address health literacy and language barriers. “We went to the Korean center that has Korean-speaking people who go out in the community, and they help translate. They’re sort of our translators”.

Participants discussed how their interpersonal relationships in the community positively impacted their approach to cancer care delivery for low-income and racially and ethnically minoritized populations. One stated, “I guess in community practice that comes along based on your interactions with [other community members]. Like, at least with my gastroenterologist, we’ve talked so often before about these patients have no real insurance; how do we tackle them? And then we discussed it with the disease management team and came up with the pathway of accepting them”. Another stated, “Someone might need something like lymphedema therapy, and they’re not in the network with the closest location, but they need to get to another part of town and they can’t because of distance because of the transportation. Well, the [community organization] provided with a grant for funding the hospital and also working with the clinic to provide rideshare to provide medical Uber rides, and that has helped a number of individuals get to appointments or be able to get the care they need”. Another stated, “It’s about who you know and how you know them. You know we go out and talk to people in the town and other advocacy groups and say, ‘here’s the problem,’ and then we can come up with the solutions together”. One stated, “We told [the surgeons] about those patients who are waiting for elective surgery related to their cancer diagnosis and how they were at risk for bad outcomes because they were already delayed, and they put them on an emergency list, and they’re operating on them”.

### 4.10. Theme 2: It’s a Calling, Not a Job

All participants noted that their passion and ties to the community facilitated the delivery of evidence-based care for populations most at risk for cancer disparities. One stated, “If you don’t help the underserved population.. well, you see the difference every day”. Another stated, “I was born here. I was actually born in this hospital, the hospital where I practice. My great grandfather was sheriff here. So, I have deep ties here, and I don’t know; it’s really tough to recruit, but I grew up in this rural setting so I always knew I wanted to come back”. Another noted, “So I finished my fellowship and started with this practice, and I asked when I was interviewing about where there was a place I could focus on [underserved populations] both in terms of patient population, demographics. and research”. One stated, “It’s more than just a job to all of us. This is what we are supposed to do”.

### 4.11. Theme 3: Experiential Training

Participants noted that they received little to no formal training in working with populations most at risk for cancer disparities. All participants agreed that the care they delivered was based on practical experience. For example, one participant described, “Unless someone is here, all the knowledge you get outside is all book knowledge. It doesn’t apply to your circumstances. The learning takes place here on [the] ground in the surroundings in this environment”. Another participant stated, “The challenges are incredibly difficult, not insurmountable, when you complete your fellowship, and even if you’ve spent time in an academic center, and you then transition to a community-based program, there is an idealistic approach. But if you read through the published information, it doesn’t help you”. Another participant reflected on how their first-hand experiences, such as witnessing racism, influenced how they approach their clinical practice. “I did not have any formal training about that, honestly. I wasn’t aware of the magnitude and the history of the United States. I knew about the racism and racial discrimination, and so on and so forth, but I did not realize to what resources are available, and the demands on the patient, and the degree this has also extended to our time now and continues in every single day on certain groups of the population, and your time constraints. I don’t think that is absolutely clear [when you move to an under-resourced setting], specifically on our Black community. But I totally understand this problem and in completely different terms and different ways”.

## 5. Discussion

In this qualitative research study, oncology clinicians described their experiences in delivering care for populations most at risk for cancer disparities, specifically low-income and racially and ethnically minoritized populations. Study findings uncovered key barriers that contribute to ongoing cancer care delivery disparities and clinician burnout. The study also revealed facilitators that can help overcome these barriers and contribute to improving evidence-based care equitably and clinician wellbeing.

Clinicians not only delivered care with a dearth of resources but also increasingly devoted time and effort to advocating for resources to facilitate evidence-based cancer care delivery for underserved populations. In 2017, the American College of Physicians position statement recommended that administrators and executives analyze and mitigate or eliminate the adverse effects of administrative tasks on clinicians [19]. Our research findings, four years after these recommendations were made, further support this need, especially in clinical settings where policies and practices may place undue burdens on understaffed and under-resourced teams. For example, a critical evaluation of the care teams’ workforce composition, such as the efficient and most effective use of social workers and navigators, can identify if the care team is appropriately staffed to address health-related social needs. The correct composition of care team members, with each team member working to the top of their licensure, not only improves workplace efficiency but also more effectively delivers care for populations with complex social and economic challenges. Studies show the value of such team compositions, specifically in the consistent delivery of high-quality care and cost savings via the elimination of time spent by clinicians on activities that may be more effectively and efficiently delivered by other members of the team [20,21,22]. Our findings reveal that the burden of team staffing evaluations was often organic and shouldered by the clinicians who were delivering the care. Such evaluations should not be the sole responsibility of the physicians delivering the care alone but also be heavily influenced and led by the administrative leaders responsible for resource allocation.

Our results overwhelmingly highlight the importance of organizational commitment at the executive level, including visible long-term actions and investment dedicated to improving the delivery of care for underserved populations. While most participants had organically created process and practice workflows to ensure that they and their teams could deliver evidence-based cancer care for low-income and racial and ethnic minority populations, participants consistently noted a lack of executive organizational commitment to such improvement efforts. Consistent with prior work documenting the impact of conflicting policies and practices on moral distress among clinicians [23,24], our study revealed that organizational barriers, including a lack of devoted resources and investment, often resulted in conflict with professional commitment to ensuring equity in care delivery for underserved populations. It is well known that such conflict and moral distress can greatly contribute to burnout and impact the wellbeing of cancer care clinical team members [23,25,26]. Clinicians in our study reported that their under-resourced teams, in addition to continually having to advocate for resource allocation, diverted their time and focus from more clinically important activities, which was linked with stress and burnout for them as well as added stress due to their concern for the wellbeing of their team members.

Interpersonal and professional relationships with other clinicians in the community and community-based organizations were consistently identified as a major facilitator for evidence-based cancer care delivery among low-income and racial and ethnic minority populations. This organic network of clinicians and community-based organizations served as an “underground railroad” network with a shared vision that collectively mobilized resources and directed them to ensure the delivery of evidence-based cancer care. Such networks of community and local engagement are recognized by the Institute of Medicine and others as the key component of addressing cancer disparities at the local level [27,28,29]. As noted in organizational behavior literature, community capacity is crucial to achieving change and involves building both political and social capital within and outside of the community [30]. As demonstrated in other studies evaluating facilitators for health promotion, skills in community capacity building and collective efficacy [31], or shared values and norms for the common good, were identified in this study as a key component of care delivery in resource-limited settings and were skills attained only through experiential learning, as participants highlighted. A more formalized process to collectively solve problems relating to promoting health equity should be developed and led by the organization as a whole. Such organizational shared vision, leadership, voice, and power can expand the reach and scale of organically derived networks and is necessary to prioritize resources and infrastructure necessary to deliver equitable evidence-based cancer care at the clinic, system, and community levels.

Burnout was identified as an ongoing, critical issue due to the many barriers revealed in this study. Administrative tasks, the constant need to prove the value of the resources for ensuring care delivery for underserved populations, and under-resourced teams contributed to ongoing wellbeing concerns among clinicians. While passion for care delivery among underserved populations was noted as a facilitator for ensuring equitable care delivery, such professional fulfillment also contributed to burnout, especially when systemic resource limitations or the complex social and emotional needs of patients inhibited evidence-based care delivery. Concern for other members of the team and their wellbeing also contributed heavily to the mental load and wellbeing of clinicians.

Modifiable barriers identified in this study are ripe for potential solutions that ASCO and other organizations may implement. These solutions, depicted in Table 4 and derived from key barriers identified in this first phase of the SPUS Task Force, will be evaluated in a national survey distributed to oncology clinicians across the United States to determine which solutions should be considered and prioritized by the national society. Such findings will help to move from description to action in the support of oncology clinicians caring for populations most at risk for cancer disparities.

Our study had several limitations. First, our method of identifying participants relied on a convenience sample. Due to the lack of available data regarding clinicians who predominantly deliver cancer care for low-income and racial and ethnic minority populations, the current study utilized non-random, purposive sampling (based on the geographical location of participants), relying on lists generated by members of the Task Force and the HEOC to identify potential participants for the study. As such, the sampling strategy may bias the results. Second, the study was conducted in 2020, during the early phase of the COVID-19 pandemic, when many healthcare settings and community support organizations experienced unprecedented challenges in delivering care and support for patients. The response rate to our requests for interviews was around 50%, with the primary reason for non-participation given as “time limitations”. It is unclear, if these time limitations were due to limitations of routine practice or were due to additional burdens attributable to the COVID-19 pandemic. Either way, it is possible that our study sample was biased towards clinicians within more resourced settings or who have personal characteristics that make them more likely to respond. Third, no participants self-identified their ethnicity as Hispanic or Latino/a/x or a race other than the ones we listed. Due to the small proportion of oncology clinicians nationally with these self-identified characteristics, it is unknown whether experiences may differ. Finally, our study was conducted only among US-based clinicians and in English, limiting findings. Strengths of our study, despite these limitations, were that we had an inclusive, diverse sample of respondents with varied experiences delivering cancer care for low-income and racial and ethnic minority populations, including perspectives of care delivery for unregistered migrants, refugees, and similar groups across the United States. Additionally, we achieved thematic saturation across our interviews, allowing for confidence in the representation of our sample and the generalizability of our findings.

## 6. Conclusions

The current study provides key insights into the barriers and facilitators faced by clinical oncologists delivering cancer care for low income and racial and ethnic minority populations. Clinicians identified modifiable barriers, most notably their difficulty in convincing executives and administrators to understand the complexities of care for underserved patients and the need for additional resources. While clinicians are adept at working with what they have and building workflows to ensure the best possible care for their patients, the numerous day-to-day challenges contribute to moral distress and burnout. Collective efficacy and community capacity across multiple levels are important and necessary facilitators for care delivery in under-resourced settings. This work lays the foundation for understanding how and why clinicians work with underserved populations and provides insight for future actions to support clinicians and improve cancer care for these populations.

## Figures and Tables

**Table 1 cancers-15-03311-t001:** Respondent Demographics (N = 12).

Characteristic	N (%)
**Self-Identified Gender ^a^**	
Female	6 (50)
Male	6 (50)
**Respondent Ethnicity**	
Hispanic or Latino	0 (0)
**Respondent Race ^a^**	
Asian American Native Hawaiian and other Pacific Islanders	2 (17)
Black or African American	4 (33)
Non-Hispanic White	5 (42)
No Response	1 (8)
**Years since Completion of Terminal Degree**	
1–5 years	1 (8)
6–10 years	4 (33)
11–20 years	2 (17)
>20 years	5 (43)
**Percentage of Clinical Time Caring for Underserved**	
<25%	1 (8)
25–50%	7 (58)
51–75%	1 (8)
>75%	1 (8)
No Response	2 (17)
**Lives in the Community That They Deliver Care**	
Yes	8 (67)
No	2 (17)
No Response	2 (17)
**Primary Practice Setting**	
Academic Medical Center or University	4 (33)
Physician-owned practice or group	2 (17)
Hospital or health-system-owned practice, group, or department	5 (42)
Other	1 (8)
**Number of FTE Oncology Providers and Subspecialists**	
0–10	4 (33)
11–20	3 (25)
21–30	0 (0)
31–40	2 (17)
40–50	0 (0)
>50	1 (8)
No Response	2 (17)
**Patient Population Reflective of the Community**	
Yes	10 (83)
No	2 (17)
**Insurance Status of Patient Population ^b^**	
Private/Commercial	10%
Medicare/Medicare Advantage	40%
Medicaid	25%
VA/other government	7%
No Insurance or Other	18%
**Race and Ethnicity of Patient Population ^b^**	
American Indian/Alaska Native	1%
Asian American Native Hawaiian and other Pacific Islanders	15%
Black or African American	23%
Hispanic or Latino	19%
Middle Eastern/North African	2%
Non-Hispanic White	32%
Multiracial/Multiethnic	8%

^a^ Additional response options were offered beyond binary male or female gender identities (e.g., non-binary) and racial categories listed (e.g., American Indian, Alaska Native, Native Hawaiian, and Multiple Races, among other categories), but no respondents self-identified with these expanded categories. ^b^ Responses based on estimates provided by participants and aggregated across participants.

**Table 2 cancers-15-03311-t002:** Qualitative Themes and Subthemes Associated with Barriers to Care Delivery Identified by Clinicians.

**Theme 1: Lack of recognition among executive leadership regarding resources needed to deliver equitable cancer care for populations most at risk for disparities**
So me talking to the administration, they don’t get it. They don’t understand what it takes to run a cancer center, support a cancer center.
At least we are aware of the fact that we are short-staffed. We need to hire a dedicated social worker. If we can find the funds, it would be great. So we have to survive with whatever we have. We are working to improve it. We speak closely to the administration, but we got to wait until we see, honestly.
Because our nurse navigator or social worker are very well-versed in these areas, I depend or rely on them to assist with what resources within the community are available. I have to say that it’s still hard. You need administrative buy-in for this as well.
But what we have been able to demonstrate to [executive leadership] is that there is actual value to be had, return on whatever upfront investment they made. And we always assure [executive leadership] that we will be good stewards of our resources. We wouldn’t ask for stuff without demonstrating upfront the value that that upfront investment was going to make. So that has allowed us to bring in people who are able to do things like understand who needs what service, what support we can deliver beyond the direct cancer-related care and decision-making.
Challenge that I deal with every day is fragmentation of care. And so to try and get all these pieces coordinated in a timely fashion, real challenge. So we spend a lot of time and staff effort for that, which the administration here doesn’t understand.
We don’t even have a dedicated social worker here. The patient navigator, she does the social work. Whenever we are faced with transportation challenges or meals on wheels and all that, we ask her if the patient is eligible for those programs. She goes above and beyond to help those patients.
So we don’t need temples. We need marketplaces. We need places where people can go in and transact. I go in, I have a need, I can be sure that I’m going to get it because, Oh, I am welcome, and the services are there. And yes, the goods are high-quality. Why do I know that? Because the people who run the marketplace are truly held accountable for making sure that the goods they provide for me to purchase are high quality goods. Right now, that’s not the way we have [our clinics] set up.
Basically, it all boils down to having enough resources to make these things to make these changes that are unique to this population. [Other places] are able to assign additional resources to make the hospital function better. How does it feel to have that luxury?
We are asked to keep showing value. For example, in the past, a guy doesn’t have insurance, so can’t come in. We have been able to negotiate that away with the healthcare system, where there is this idea of accepting the loss lead, if you will, that allows people to come in understanding that [the health system and clinic] is efficient enough to be able to swallow whatever minor up-front expenses there are. But it is a challenge [for them] to see the big picture, but that big picture allows you to understand that your revenue is not based on this single encounter that looks like a loss.
I would bet that what happens in other practices like this is that significant corners are cut just because there’s no resources to support providing navigation services, and they don’t have the nonprofit organization to provide those services; the quality is going to suffer tremendously in this patient population when they don’t have access to the navigation services.
The main carrot that you’d have to figure out is how do you make providing cancer care in these communities more lucrative so that larger organizations do want to branch out and provide care in those communities and fill the gap that now is being served by most single doctor shops and probably not filled that well.
So all of these [approaches for improving care] we have done in such a rigorous way that I have been able to go back to my senior administrators and show them data that has encouraged them to invest more resources into these programs as we’re building them.
Because of course, you don’t want to get administrators in the business of driving their healthcare systems bankrupt, but you also don’t want to be in the business of excluding people because of who they are or what they are. And so that has created a real opportunity to look at care delivery programs and how we can optimize them to do both: to be both financially solvent while also opening access to those who truly need it. It’s dependent on us, though, to have to make that argument.
The inadequacy of things we have around here.. I don’t internalize that [the leadership] want to go and kick the dog. But we are always looking for funding for our needs to do this work.
I want to say we are able to overcome the disparities because [they] are in a position to invest in the resources that are needed to help our patients. In other words, the challenge is that patients, they may not have the healthcare literacy to follow through the complicated intersections of an oncology patient; we don’t have enough navigators to help them. We don’t have enough social workers to be assigned to the patient. This is tough for me personally.
**Theme 2: Evidence-based care delivery is challenging due to patients’ unmet complicated social and economic needs.**
I’ve learned that in my time here, that it’s meeting the needs of a patient more so than their cancer.
People who don’t have access to resources will avoid medical care just because they know it’s going to entail something; you know, if they’re going to have to pay money or get people to help them get somewhere. So, it’s almost as if, so I know that people may delay medical here until they just can’t avoid it, and that, I think, is pretty common.
For the past several decades, we have made assumptions that we have stereotype patients that are “noncompliant”. Not really taking the time to understand what their obstacles might be. Things that we take for granted that we really would never understand. It takes time to get to know your patients [and their needs.]
It’s a mixed bag. One of the barriers, I think, with the health literacy is a lot of our uninsured patients, their first language.. They don’t speak much English at all, so that’s been a complicating factor… there is significant issues with health literacy.
There are many services patients actually don’t have access to… many primary needs, like transportation; patients don’t have appropriate phones, they didn’t have appropriate Wi-Fi internet service, they did not have email set up to begin with. People who may not have had resources to begin with, and then, when the pandemic hit, being left behind in this big innovative technology shift because they don’t have the basics.
[My patient] told me that she missed one appointment, and she was fined because the missing appointment. I think it was $66, $65, something like that. She missed the appointment because they couldn’t have transportation. So there is a pull-and-push dynamic. So for these different reasons, the proportion of patients who are, I would say, live in poverty, are of certain racial groups, or live away from the Med Center are the ones who come to us because we always are accommodating them.
Language is a big barrier. Transportation has been a big barrier, but not at my current institution. They actually have a really good setup where we’ll arrange Ubers for any patient that needs transportation issues. So that’s been really helpful at this hospital that I didn’t have at other hospitals. So I’ve even been able to get them in a lot easier that way, but yeah, job issues. I’ve had patients who just refuse to stop working and would miss several appointments because they prioritize their jobs. They needed money over coming in for treatment. That’s been a barrier sometimes.
**Theme 3: Burnout is prevalent among clinical providers due to lack of resources, time, and effort spent advocating for resources and the social and emotional challenges patients face in these settings.**
The community groups are providing transportation because of the concerns about safety, and family members may have their own concerns. So, the system is stressed and that means the people are stressed, and the doctors are stressed, too.
My population has, as it does for cancer in general, skews older, and so patients are reliant on others to be able to get them back and forth for appointments. There is a real health literacy issue. Patients don’t want to see the doctor, and so I see patients when they tend to be very far advanced at the time of diagnosis. They don’t come in for screening and when they have a cancer, they don’t come in until they absolutely have to. So when I was starting as the only full-time oncologist here in this region, I was really worried. How do I be on call 24 h a day every day? And so yeah, lots of challenges in this very rural, very isolated place.
It’s frustrating when the administration won’t listen to me but they’ll listen to outside consultants. So yeah, the same things that I was saying and making the case for, the consultants were like, “Yeah, you need someone to help coordinate these new patients. You really need a position; some of the other aspects of running a cancer center”. But you had to get someone else to come in and say it. So yeah, story of my life. This contributes to my personal struggles here in a big way.
I’m very passionate…and I don’t give up, which means I will push and push until I can’t.
Admittedly, it breaks my heart, but we end up turning away patients that don’t have insurance. We just can’t financially treat them. The margins are too small to have uninsured patients and write off. We just can’t do it; a small center can’t do it. There’s just not good ways around that, so we directed the patient to a larger center. It breaks my heart.
But yeah, there are real issues and I can think of a case recently; man was essentially living in a trailer, didn’t have running water, didn’t have heat, didn’t have any family support. We don’t have good options in the community to treat patients like that..living in a situation where we couldn’t safely treat him..and so we transitioned to more of a palliative plan of care, and that’s just the reality. It’s a hard reality for me to see this day in and day out.
You know, I think I struggled with that significantly. And also with the fact that, socially, some [patients] are not so supported at home. And sometimes I feel like there’s very little I can do to change that. I mean, I can give them the chemo. I can watch them for toxicities, but I really can’t take care of so many of the stressors they have at home to make them feel better through their journey. I see a lot of metastatic end-stage patients that with different spectrums, and it’s discouraging.
The professional aspects related to taking care of this population have been quite a challenge. It’s often navigating a patchwork of care that is really challenging for even a medically literate person with means to navigate. This is what has really bothered me the most.
When you lose that depth of commitment from the team, especially when trying to do disparities work that’s important, but in terms of burnout, I am struggling with figuring out a program that can be put into place.
I think being able to get away is important. But it’s not always possible. You run into danger when you have just one oncologist or one advanced practice provider. In the last two weeks, I’ve received the equivalent of four months of referrals, and we are overwhelmed. Patients that are very sick lots of nonmedical needs and medical needs, and my schedule’s full, and it’s shocking. It feels like it doesn’t end. We struggled through COVID, and yeah, the pandemic’s getting better, and now we’re getting slammed when we’re exhausted. So yeah, it just doesn’t end.
More than often, most of us will put 125% effort in doing it. I wonder how much of that can also play into burnout. When people don’t feel appreciated, they tend to burn out or they get bored doing the same thing.
All the frustrations come out of that fact that we need more resources.
So you add all those levels, and you realize that your life is meaningless now. I come here, work hard, go home, work hard. Wake up. Work hard. Come back to work the same story, and I am not doing what I have signed up for, so from my perspective, I think addressing all these three issues becomes extremely important; at least two of them, I would say: the medical records and the mission. We are going to lose our people.
There’s definitely frustration when you want to see a patient, especially in the hospital, and you’d want to start treatment, but you just can’t, and you know that’s delaying their care…that can be a huge area of frustration.
Working painfully over decades what I came around to recognize was I lost my children’s childhood entirely. So that can make for burnout. I struggled for a few years here. Yeah, professionally very rewarding. We’ve been able to do this and that, but it was tough.
The work is time-consuming. It is demanding. With all the rewards that come from being able to serve, we can get blinded to the opportunity cost at the personal level. And families do suffer. The families of healthcare workers suffer.
You want to treat that patient and either the health literacy or cultural issues or whatnot, there’s a whole variety of reasons why the patient cannot get that treatment. You wonder what else you could have done or what not. I think that takes up a lot of time and can be tiring.
So, the cancer center is only a part of the main hospital, right. It’s not the predominant part, so, you know, burnout affects the whole hospital, right. It just doesn’t affect the cancer physicians.
Certainly, there are situations where I feel frustrated by maybe what’s happened with the patients before they come into our system or getting certain kind of services that are not allowed by insurance.
Here, you know if you order a test, you don’t know whether it’s going to be done/not done. Are you going to struggle getting that patient a referral for a gastroenterologist? Because no one takes that uninsured charity patient. Now it’s unfortunately a lot at the forefront because you know you will not be able to take care of the patient the same way. You’re going to struggle through other aspects of care.

**Table 3 cancers-15-03311-t003:** Qualitative Themes Associated with Facilitators of Care Delivery as Described by Clinicians.

**Theme 1: Local connections with community partners and foundations**
The community organizations not only do they translate and interpret, what they do is, if I say, “Okay, you need to see a GI and surgeon in the next week because this is really important,” I message the interpreter, and in addition to the whole explaining to the patient, so that she follows up on it on the backend from the community center.
In my time here, I’ve learned where to send patients, where to do different things, and who to send them to.
[Community partners] bring in navigators they funded for several years.
One community organization has nurse navigators trained to provide [infusion] services at [community-based] locations for things that are limited and treatments that we feel are safe to give closer to home.
We also have a liaison in the community who works with the hospital and our clinic to set up opportunities when people would travel from afar to stay at local hotels. We are able to get a substantial amount of gift cards that help with not just groceries but bill pay because certain agencies can do one aspect but maybe can’t meet the other. So we have gotten creative about how we can help pay for a utility bill or a person’s groceries.
The [community organizations and community members] go out of their way to help get patients here, and we have comfort funds to help patients get care who have difficulty getting care.
Our communities have resources to help cancer patients that help to educate our patients.
So, I had a patient that I saw who needed palliative treatment. And she said, ‘Well, I don’t have a car, and nobody in my family has a car.’ So, we looked in the community, and there was someone in the community who could drive her. And then the next day we get a call like, ‘We can’t. The road ends before our house.’ So, the person in the freestanding center says, ‘I know someone who works in the Department of Transportation,’ and she called them up, and they had the road cleared. So, they actually made a road within a two-day period so somebody can drive and bring this patient in for treatment. It’s pretty amazing. So, we have our challenges, but we have help as well.
I have an intimate, I would say, interaction with leaders in the community. Also, I was introduced to the ins and outs and what happens to folks who don’t have the means and how we can support them. We distribute a lot of those grants that we get to local communities to support transportation, sometimes meal vouchers, for those who need to travel and stay away from their homes so they cannot go back and eat that same day.
We have these workarounds. Screening colonoscopies is definitely a struggle. We’ve been getting a few physicians who’ve been kind enough to say, ‘Listen, I would do these without a question. I will not charge them, and I will do them in my private office.’
We have multiple connections out there in the community to help empower and engage the lay people.
**Theme 2: It’s a calling, not a job**
This for me is not just a choice, it’s a duty; I have to do it.
The work I do is definitely more self-motivation. I have always been interested in health disparities, and so I knew early on this is what I was going to do.
The self-interest makes you wake up early in the morning and stay up late at night doing stuff to where the stuff you do is aligned with a greater purpose, mission that you subscribe to. That’s it for me.
This job was a combination of serendipity and recognized opportunity.
There are many of us out there that want to help. We just want to be planted in the right direction. We want to be a part of change. We want to be a part of implementation. We want to be able to take the conversation to the next level, we really do.
My overall goal is to provide underserved patients with a comprehensive team with differing skillsets to help them navigate and overcome the challenges that they face. I educate myself all the time with regards to no shows and this and that, that dig deeper.. dig a little deeper to better understand where our patients are coming from.
I think we have everything we need: native intelligence, interest. And then we just have to apply ourselves diligently to whatever challenge we’re grappling with.
I left another organization for the opportunity to care for a more diverse patient population.
I’m the best chance [the patients who are undeserved] got, and I could try to pass this off to someone else but who? I can’t necessarily get them into a primary care provider, or I could get them to a primary care provider but someone who’s not able to manage a super complicated situation like this. It’s on me.
My goal was that patients wouldn’t sacrifice anything by being treated here, that they would get the standard of care here that they would get anywhere else. I can confidently say that that’s true, but through a lot of trial and error.
People need to understand that without job, without housing, without transportation, without healthcare insurance, without health literacy, without all these components, now what you call it collectively the social determinants of health, without tackling all of them at once, we will not be able to provide our patients with the appropriate care that they deserve. Trying to change that culture that wants to ignore all these elements, thinking that it’s your responsibility. No. It is our responsibility.
**Theme 3: Experiential Training Was Key to Delivering Care for Underserved Populations**
The informal training was during my residency. My residency gave me a good foundation going into my oncology fellowship to think maybe how I should think about questions and people differently.
We didn’t really get any instruction on [how to deliver this care]. Then certainly, in practice, out at meetings and stuff, it’s nothing that I focused on. I imagine there probably are some lectures and access to [formal lectures], but it’s nothing that I sought out.
I learned by doing. It taught me, at least, right off the bat that you have to handle these patients differently.
I did my fellowship at an institution where I saw few underrepresented groups. It wasn’t until I started practicing that I learned how to do this.
When you complete your fellowship and then you transition to a community-based program, there is this idealistic approach that you will have all the resources available. But the demands of the patient population and your time constraints and what you need to do to make sure people get care… well I don’t think that that is absolutely clear [in fellowship or other formal training].
Early on in my residency, I learned that if the patient was sitting in front of me, I would actually pick up the phone and schedule the scans and the blood work and set it all up before they left. It was almost like … because there was no real secretary to help with all of that, right? And so, I think early on, I think I learned that there’s a population that, unless you set up very clearly all the expectations and line up things, the chances are they may not be able to follow through.

**Table 4 cancers-15-03311-t004:** Potential Solutions Derived from Findings.

Strategies directed to health care executive leaders regarding value of cancer health equity, including education and awareness campaigns and webinars regarding resource requirements for high-quality equitable cancer care delivery. Identify and disseminate best practices regarding evidence-based cancer care delivery in low- and under-resourced settings. Strategies directed to address burnout and wellness among clinicians delivering cancer care for underserved populations, collaboration, and the establishment of a task force directed to this growing concern. Networking opportunities among clinicians delivering cancer care for underserved populations. Highlight community capacity building and collective efficacy strategies at national meetings, webinars, and via collaboration locally with State-Affiliate leaders. Development of experiential training opportunities for medical students, residents, and fellows interested in oncology across disciplines, including medicine, nursing, public health, pharmacy, social work, among others, to immerse them in hands-on learning through rotations, internships, and mentorship.

## Data Availability

Data Availability Statement: All data supporting the findings of this study are available within the article, Appendix A, and from the corresponding author upon reasonable request.

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
