# Peer review of "Assessing the Needs of Those Who Serve the Underserved: A Qualitative Study among US Oncology Clinicians"

_cancers, 2023, doi:10.3390/cancers15133311_

Round 1

Reviewer 1 Report

From a biostats/clinical epidemiology point of view, I have no major concerns about this very interesting qualitative study coming from a large group of ASCO Health Equity Committee’s Services for the Underserved  (deriving from the Health Equity and Outcomes Committee-HEOC) colleagues.

An original interview protocol has been submitted as a supplement, what about its potential validation!? Have you planned different translations into other languages!?

Why have you invited only 24 clinicians!? I do suppose that many more could haven taken part to this survey that investigated underserved population in clinical oncology

Moreover, how do you believe to increase the rate of participation to similar surveys, since this time it was only 50%!?

Please, mind several undefined abbreviations (FTE, VA, NP and so on)

Author Response

Thank you for you for reviewing our manuscript and providing us with the opportunity to improve upon our work. We have carefully reviewed the comments and have significantly revised the manuscript accordingly. Our responses are given in a point-by-point manner below. Changes to the manuscript are shown in two versions, one using the track changes and another using a clean version with all changes accepted.

Reviewer 1.

From a biostats/clinical epidemiology point of view, I have no major concerns about this very interesting qualitative study coming from a large group of ASCO Health Equity Committee’s Services for the Underserved (deriving from the Health Equity and Outcomes Committee-HEOC) colleagues.

            Thank you for this kind comment.

An original interview protocol has been submitted as a supplement, what about its potential validation!? Have you planned different translations into other languages!?

Thank you for this comment. In qualitative research, we followed the discussion by Lincoln and Guba (1985) regarding trustworthiness. Unlike quantatitive research, in qualitative research, we cannot apply validation to the interview questions. However, we did much upfront work in preparing the interviewing guide. For example, prior to creating the interview guide, we invited expert opinions from prior chairs of the Health Equity Committee in ASCO as well as prior members of the Health Equity Committee and current members of the Health Equity Committee to discuss content of the interview guide. We then asked these same experts to provide feedback on the interview guides once drafted. We then added all the feedback and in conducting the analysis, we used triangulation to ensure trustworthiness. We added this to the Methods section. The methods now state, “The interviews were conducted by video conferencing through a web-based platform between March 1, 2021, and April 30, 2021, by investigators (JM, KS) using a semi-structured interview guide developed by members of the Task Force and created in discussion with members from the HEC (Supplemental Appendix). The interview questions were developed using the Socio-Ecological Framework [11].”

Translation of the interview guide would be a great consideration for a future study. In this study, our focus was on identifying barriers and facilitators among US-based clinicians who are all English speaking. We added this to the limitations section. Specifically, we now state, “Finally, our study was conducted only among US-based clinicians and in English.”

Why have you invited only 24 clinicians!? I do suppose that many more could haven taken part to this survey that investigated underserved population in clinical oncology. Moreover, how do you believe to increase the rate of participation to similar surveys, since this time it was only 50%!?

Thank you for this comment. This was a 1-hour long interview with open ended questions that were semi-structured. As we knew that we needed approximately 12-15 participants to reach thematic saturation, we reached out to 24 clinicians. We reached thematic saturation at 10 interviews but continued with the 12 as the other 2 interviews had already been scheduled and the clinicians really wished to participate. We anticipated a response rate of close to 25% given prior work conducted by ASCO as well as the timing of the study during the height of the COVID-19 pandemic. Furthermore, given that we were targeting clinicians who were serving underserved populations, these clinicians were disproportionately impacted during this time by the pandemic and therefore this may also explain the 50% response rate. We revised our methods which now state, “A total of 12-15 participants were anticipated to reach thematic saturation. However, as this study was conducted during the height of the COVID-19 pandemic and given the time limitations of clinicians targeted for study participation, the Task Force sent email invitations inviting 24 clinicians to participate in 30-60 minute, virtual, semi-structured interviews.”

Please, mind several undefined abbreviations (FTE, VA, NP and so on)

Thank you for this comment. We revised the manuscript to address this comment.

Reviewer 2 Report

This is an important paper reporting findings of a study by ASCO’s “Supporting Providers Serving the Underserved (SPUS)” Task Force. It reports qualitative findings about the barriers and facilitators to equitable cancer care perceived by clinicians serving the underserved. As such, it gives voice to clinicians struggling amidst complex systemic issues and its publication will be an affirming step in support of both these clinicians and the populations they serve.

I just a have a few minor questions/edits to suggest for the authors :

1.     The rationale for the purposive sampling is well described and the method is appropriate. I am curious and suspect the readers would be too to know how many oncology clinicians were identified in all? Before purposive sampling by geography and other factors of the 24 invited to participate.

2.     Please add a reference for the thematic analysis theory/method used.

3.     Generalizability is not usually an expected attribute of qualitative methods (Leung doi 10.4103/2249-4863.161306) so this is not a limitation the authors need to focus on in line 413, the sentence could stop at bias results.

4.     Did participants share perspectives on the barriers/facilitators of treating refugees, unregistered migrants and similar groups facing marginalization or is that outside the scope of this paper? If included that strength could be added to lines 427-428.

5.     The paper could be strengthened by adding a supplement with positionality statements for each of the authors and their qualitative research experience, this will add to the robustness of the paper.

6.     I was curious about why certain demographics collected are not show in the article e.g. results for whether the population treated is reflective of where the practice is located. Can this be explained or added to the demographics table?

7.     The discussion is a little bit repetitive of the results. Are there references from other equity studies about the strengths and facilitation of having community ties? These would be helpful for readers. In addition, a box of recommendations or next steps of the SPUS task force will take in relation to these findings would enhance the utility for clinicians advocating in their local contexts and nationally.

8.     When published it would be helpful to have the tables of quotes laid out dispersed between the sections that they pertain to.

Author Response

Thank you for you for reviewing our manuscript and providing us with the opportunity to improve upon our work. We have carefully reviewed the comments and have significantly revised the manuscript accordingly. Our responses are given in a point-by-point manner below. Changes to the manuscript are shown in two versions, one using the track changes and another using a clean version with all changes accepted.

Reviewer 2.

This is an important paper reporting findings of a study by ASCO’s “Supporting Providers Serving the Underserved (SPUS)” Task Force. It reports qualitative findings about the barriers and facilitators to equitable cancer care perceived by clinicians serving the underserved. As such, it gives voice to clinicians struggling amidst complex systemic issues and its publication will be an affirming step in support of both these clinicians and the populations they serve.

Thank you for this comment.

I just a have a few minor questions/edits to suggest for the authors:

The rationale for the purposive sampling is well described and the method is appropriate. I am curious and suspect the readers would be too to know how many oncology clinicians were identified in all? Before purposive sampling by geography and other factors of the 24 invited to participate.

Thank you for this comment. The list that was generated had a total of 50 clinicians identified across the United States.  From this list, we purposively selected a random 24 based on geography to participate. We added these details to the Methods section which now states, “The SUS Task Force, in collaboration with prior and current HEC members, generated a list of 50 medical oncologists across the United States who, based on their knowledge and expertise, spent greater than 25% of their time treating or having had experience treating predominantly low-income and racial and ethnic minoritized populations with cancer in their daily clinical practice.”

Please add a reference for the thematic analysis theory/method used.

Thank you for this comment. We used methods by Braun and Clark (2006). We added this reference to the manuscript in the methods section which now states, “Analysis was performed using thematic analysis as described by Braun & Clark [12], given the limited research on the topic. The flexibility in this approach permitted an understanding of experiences of participants in the broader context of their roles delivering cancer care for underserved populations while remaining focused on the data and recognizing practical and realistic limitations [12].”

Generalizability is not usually an expected attribute of qualitative methods (Leung doi 10.4103/2249-4863.161306) so this is not a limitation the authors need to focus on in line 413, the sentence could stop at bias results.

Thank you for this comment. We appreciate the suggestion to remove this from our limitations section and removed this from the manuscript.

Did participants share perspectives on the barriers/facilitators of treating refugees, unregistered migrants and similar groups facing marginalization or is that outside the scope of this paper? If included that strength could be added to lines 427-428.

Thank you for this comment. Yes, participants’ perspectives reflected these experiences as well.  We added this as a strength in the lines suggested.

The paper could be strengthened by adding a supplement with positionality statements for each of the authors and their qualitative research experience, this will add to the robustness of the paper.

Thank you for this comment. We added a supplement with positionality statements. In addition, we added descriptions of the investigators to the Methods section regarding qualitative research experience. Specifically, we now state, “Three investigators with experience in qualitative methodology and backgrounds in public health, anthropology, oncology, psychology, ethics, social work, and health services research (MP, LH, FH) read samples of the transcribed text to get a broad understanding of the data content.”

I was curious about why certain demographics collected are not show in the article e.g. results for whether the population treated is reflective of where the practice is located. Can this be explained or added to the demographics table?

Thank you for this suggestion. We added these characteristics to the manuscript from the demographics table. “Most (n=8; 67%) lived in the community they delivered care and were employed by a hospital or health system owned practice, group or department (n=5; 42%) with 10 or less Full-Time Equivalent oncology clinicians and subspecialists (n=4; 33%).  Most (n=10; 83%) reported that their patient population was reflective of the broader community in which they practiced and reported that the most of their patients were insured by public insurance, either Medicare or Medicare Advantage (35%) or Medicaid (30%) with no insurance or other forms of payment (such as charity care) representing 18% of their patient population. Most participants reported that Non-Hispanic White patients represented 32% of their population followed by Black or African American (23%), Hispanic or Latino (19%) and Asian American Native Hawaiian or other Pacific Islander (15%).”

The discussion is a little bit repetitive of the results. Are there references from other equity studies about the strengths and facilitation of having community ties? These would be helpful for readers. In addition, a box of recommendations or next steps of the SPUS task force will take in relation to these findings would enhance the utility for clinicians advocating in their local contexts and nationally.

Thank you for this comment. We revised the discussion section regarding the strengths of community ties and added additional references. We also added a Table 3 that has recommendations regarding next steps for the SUS Task Force.  

The Discussion section regarding community ties is now rewritten and states, “Interpersonal and professional relationships with other clinicians in the community and community-based organizations was identified consistently as a major facilitator for evidence-based cancer care delivery among low-income and racial and ethnic minoritized populations.  This organic network of clinicians and community-based organizations served as an “underground railroad” network with a shared vision that collectively mobilized resources and directed it to ensure delivery of evidence-based cancer care.  Such networks of community and local engagement are recognized by the Institute of Medicine and others as necessary key component of addressing cancer disparities at the local level [27-29].   As noted in organizational behavior literature, community capacity is crucial to achieving change and involves building both political and social capital within and outside of the community [30]. As demonstrated in other studies evaluating facilitators for health promotion, such skills in community capacity building and collective efficacy [31], or shared values and norms for the common good, were identified in this study as a key component care delivery in resource-limited settings and were skills attained only through experiential learning as participants highlighted.  A more formalized process to collectively solve problems relating to promoting health equity should be developed and led by the organization as a whole. Such organizational shared vision, leadership, voice, and power can expand the reach and scale organically derived networks and is necessary to prioritize resources and infrastructure necessary in delivering equitable evidence-based cancer care at the clinic, system, and community level.”

When published it would be helpful to have the tables of quotes laid out dispersed between the sections that they pertain to.

Thank you. We will ask editors to include the table quotes interspersed with the sections that they pertain to.

Reviewer 3 Report

Dear authors

I would like to thank you for giving me the opportunity to review the manuscript entitled “Assessing the Needs of Those Who Serve the Underserved: A Qualitative Study among US Oncology Clinicians”. This study suffers from methodological issues that may influence the credibility, transferability, dependability, and confirmability of the findings. Data analysis was not conducted to develop themes well and with appropriate abstraction. At this time, I could not have a more positive opinion of your study. I have some suggestions to improve the presentation of your work. My comments are as follows:

Abstract

- The keywords should be added.

Introduction

- Why was it necessary to conduct a qualitative study? The justification for conducting the study in a qualitative way has not been stated

- A more extensive literature review on the subject under study is needed

Methods

-  Please add the qualitative approach used in this study.

- Discuss researcher characteristics and reflexivity - Researchers’ characteristics that may influence the research, including personal attributes, qualifications/experience, relationship with participants, assumptions, and/or presuppositions; potential or actual interaction between researchers’ characteristics and the research questions, approach, methods, results, and/or transferability

- Criteria for deciding when no further sampling was necessary (e.g., sampling saturation) should be discussed.

- Ethical issues pertaining to research - Documentation of approval by an appropriate ethics review board and participant consent, or explanation for lack thereof; other confidentiality and data security issues

- What techniques to enhance trustworthiness were used? Techniques to enhance the trustworthiness and credibility of data analysis (e.g., member checking, audit trail, triangulation)

Results

- Data analysis seems superficial. Themes and sub-themes are very long, in qualitative studies, researchers should move from objectivity to abstraction and use their creativity to find shorter themes (not sentences) for their findings. My suggestion is to consult a researcher familiar with qualitative research to improve the research methodology and data analysis

It should be carefully corrected for spelling and punctuation errors

Author Response

Thank you for you for reviewing our manuscript and providing us with the opportunity to improve upon our work. We have carefully reviewed the comments and have significantly revised the manuscript accordingly. Our responses are given in a point-by-point manner below. Changes to the manuscript are shown in two versions, one using the track changes and another using a clean version with all changes accepted.

Reviewer 3

I would like to thank you for giving me the opportunity to review the manuscript entitled “Assessing the Needs of Those Who Serve the Underserved: A Qualitative Study among US Oncology Clinicians”. This study suffers from methodological issues that may influence the credibility, transferability, dependability, and confirmability of the findings. Data analysis was not conducted to develop themes well and with appropriate abstraction. At this time, I could not have a more positive opinion of your study. I have some suggestions to improve the presentation of your work. My comments are as follows.

Thank you for this comment. We appreciate your comments to try to improve upon this work.

Abstract  

The keywords should be added.

                        Thank you for this comment. We added key words to the abstract.

Introduction

Why was it necessary to conduct a qualitative study? The justification for conducting the study in a qualitative way has not been stated

Thank you for this comment. As there is limited work to date from the opinions of the clinicians on the Health Equity Committee, it was unclear what the main barriers we needed to address in the development of a larger survey tool. There was much debate about what the needs may be and not much data about how to create a survey tool that focused on the main priority areas identified by this qualitative study. The interviews were necessary to uncover themes that would be the main focus of the study and to ascertain potential solutions that we could ask in the survey that could be prepopulated. We revised the Introduction to add this rationale which now states, “As a first step, the SUS Task Force conducted this qualitative study given the limited data available to explore barriers and facilitators to equitable cancer care delivery from the perspective of clinicians. The qualitative study would then inform a quantitative survey that would be distributed as a second step among oncology clinicians nationally to generate data on a national level that could inform recommendations and solutions.”

A more extensive literature review on the subject under study is needed

Thank for this comment. We conducted a literature review and added references regarding this subject. Most of these studies were not US-based and focused on cancer screening. The Introduction section now states, “While studies have addressed the unmet needs among clinicians in the provision of cancer screening [5-7], and studies examining the multidisciplinary care for patients outside the United States [8], limited data exists regarding barriers and facilitators that oncology clinicians within the United States experience in delivering cancer care among populations most at-risk for cancer disparities.”

Methods

Please add the qualitative approach used in this study.

Thank you. We used the approach outlined by Braun & Clark. We added this citation to the paper and also revised the Methods which now state, “Analysis was performed using thematic analysis as described by Braun & Clark [12], given the limited research on the topic. The flexibility in this approach permitted an understanding of experiences of participants in the broader context of their roles delivering cancer care for underserved populations while remaining focused on the data and recognizing practical and realistic limitations [12].”

Discuss researcher characteristics and reflexivity - Researchers’ characteristics that may influence the research, including personal attributes, qualifications/experience, relationship with participants, assumptions, and/or presuppositions; potential or actual interaction between researchers’ characteristics and the research questions, approach, methods, results, and/or transferability

Thank you for this comment. We used collaborative reflexivity approach in which we held team-reflexive discussions. During these meetings, each team member engaged in reflective discussions to answer personal reflexive questions as noted by Barry et al. (1999). These questions were posed to help team members reflect on how their experiences shaped their participation in the project. Specifically the questions asked team members to reflect on the following: What experiences have I had with qualitative research? What is my orientation to qualitative research? What results do I expect to come out of this project? What theories do I tend to favor while analyzing data? What is my stake in the research? What do I hope to get out of it? What are my fears?

We revised the Methods section which now state, “Two non-clinical members of the team experienced in qualitative methodology, interviewing techniques, and public health (JK and KS) conducted all interviews to limit impact of clinician-researcher assumptions and experiences on participants’ responses. There were no prior relationships or interactions between any of the research team members and the participants.” 

In addition, we added the following paragraph to the Methods which state,  “Discussions were held between coders and the research team throughout the analysis phase to reflect on how personal experiences were used to attach meaning to themes and if assigned meanings resonated across team members from different clinical and research backgrounds. The research team held reflexive team discussions to ensure meanings assigned to themes were consistently agreed upon by all team members as described by Barry et al [16]. This process allowed the research team to identify personal paradigms regarding research involving underserved populations, express orientation to qualitative research, negotiate a research methodology aligned with the aims of the study, and work to reduce bias in interpretation. Triangulation was applied to compare data gathered from discussions with experts held with other HEC members prior to study start and those gathered from the semi-structured interview participants [17]. Analytic memos, field notes, codebook, coding rules, and meeting notes were kept as an audit trail for dependability and confirmability [18].”

Criteria for deciding when no further sampling was necessary (e.g., sampling saturation) should be discussed.

Thank you for this comment. We revised the Methods section to add this information which now states, “Thematic analysis of 1,582 unique quotations was conducted and codes were sorted into broader themes to ensure that data with themes were consistent and that themes were distinct and fine-tuned until thematic saturation was reached, defined as a point when no new additional information was found in the data [12-15].”

Ethical issues pertaining to research - Documentation of approval by an appropriate ethics review board and participant consent, or explanation for lack thereof; other confidentiality and data security issues

Thank you for this suggestion. In the Methods, we now state, “Participants provided verbal informed consent upon initiation of the interview and completed a survey prior to the interview to define their demographic characteristics and to describe the demographic and clinical characteristics of the patient population for whom they deliver care.”

We also revised the Methods to include that identifiers were stripped from the transcriptions prior to upload into qualitative analysis software. “Interviews recorded, transcribed, stripped from potential identifiers, and imported into qualitative data management software (NVIvo).”

In our submitted manuscript in the last line in the methods section, we include the following sentence regarding appropriate review board approval. “The study was reviewed and approved by the Stanford University Institutional Review Board.”

What techniques to enhance trustworthiness were used? Techniques to enhance the trustworthiness and credibility of data analysis (e.g., member checking, audit trail, triangulation)

We used triangulation outlined by Lincoln & Guba to ensure the trustworthiness and credibility of the data. We added this information to the Methods which now state, “Triangulation was applied to compare data gathered from expert interviews with HEC members and those gathered from the semi-structured interview participants in this study. Analytic memos, field notes, codebook, coding rules, and meeting notes were kept as an audit trail for dependability and confirmability [18].”

Results

Data analysis seems superficial. Themes and sub-themes are very long, in qualitative studies, researchers should move from objectivity to abstraction and use their creativity to find shorter themes (not sentences) for their findings. My suggestion is to consult a researcher familiar with qualitative research to improve the research methodology and data analysis

Thank you for this comment. We have 2 PhD qualitative researchers and several MDs on the team with qualitative research training. The principal investigator has extensive training with qualitative methods and an honors Bachelors Degree in Cultural Anthropology. The 2 PhD qualitative researchers and MD with Anthropology experience on the team  were responsible for coding and creating the themes and sub-themes. As this was for an oncology clinician audience we have been given advice by the Health Equity Committee to create the themes into sentences for easier digestion of the information and content presented for those without qualitative research background for who the paper was written. We have now revised all of the themes and subthemes to shorter themes. We consulted an independent qualitative research analyst who is the Chair of a Department and a PhD using qualitative methodology to review methods prior to our first submission including the ASCO Research Committee which is comprised of several qualitative researchers and who also reviewed this resubmission as well.

Comments on the Quality of English Language.  It should be carefully corrected for spelling and punctuation errors.

Thank you for this comment. We have now carefully corrected for spelling and punctuation errors.

Round 2

Reviewer 3 Report

Dear authors 

Thank you for addressing my comments.